# CD44v9 Expression in Pretreatment Biopsies as a Predictor of Chemotherapy Resistance in Gastric Cancer

**DOI:** 10.3390/cancers17223657

**Published:** 2025-11-14

**Authors:** Katsuji Sawai, Kenji Koneri, Masato Tamaki, Yasuo Hirono, Takanori Goi

**Affiliations:** 1First Department of Surgery, University of Fukui, Fukui 910-1193, Japan; koneri@u-fukui.ac.jp (K.K.); m-tamaki@u-fukui.ac.jp (M.T.); tgoi@u-fukui.ac.jp (T.G.); 2Cancer Care Promotion Center, University of Fukui, Fukui 910-1193, Japan; hirono@u-fukui.ac.jp

**Keywords:** gastric cancer, neoadjuvant chemotherapy, conversion surgery, cancer stem cells, CD44v9

## Abstract

Gastric cancer remains a major global health burden, and responses to neoadjuvant chemotherapy (NAC) and conversion surgery vary owing to chemoresistance. Cancer stem cells (CSCs) exhibit self-renewal and drug resistance, influencing treatment efficacy and prognosis. Among CSC markers, CD44 variant 9 (CD44v9) is notable for its role in redox regulation and chemoresistance. This study evaluated CD44v9 expression in pretreatment biopsies from 84 patients with gastric cancer treated with NAC or conversion surgery. High CD44v9 expression (25%) was significantly associated with a poor histological response (*p* = 0.046). Kaplan–Meier analysis confirmed that a poor histological response predicted a worse prognosis (*p* = 0.045). In the multivariate Cox analysis, conversion surgery (*p* = 0.018) and poor histological response (*p* = 0.011) were independent predictors of poor prognosis. These findings suggest that CD44v9 expression levels in pretreatment biopsies may serve as a predictive biomarker of chemoresistance, guiding individualized treatment strategies.

## 1. Introduction

Gastric cancer presents a significant global health challenge, with approximately 1.1 million new diagnoses and 770,000 fatalities recorded in 2020. It is among the five most prevalent cancers and remains a major cause of cancer-related deaths globally [1]. Multimodal therapy has been adopted to improve the historically poor survival rates [2,3]. Neoadjuvant chemotherapy (NAC), administered prior to surgical resection to downstage tumors and eradicate micrometastases, has emerged as an important strategy for improving outcomes. The efficacy of NAC in improving survival has been demonstrated in the Western FLOT4 studies and the PROGIDY trial [4,5]. Despite these proven population-level benefits, individual responses to chemotherapy vary significantly. A substantial proportion of patients derive little or no benefit from preoperative chemotherapy, showing either stable disease or disease progression during treatment. Patients who are intrinsically chemoresistant not only endure the toxicity and delays of surgery without therapeutic benefit but may also miss the opportunity to receive timely alternative treatments. Conversion therapy has recently gained attention for unresectable gastric cancer with distant metastases. This strategy involves intensive chemotherapy—often combined with targeted agents or immunotherapy—aimed at downstaging metastatic disease to render surgical resection feasible. Numerous studies have indicated that patients with stage IV gastric cancer who underwent conversion surgery had better survival outcomes than those who only received chemotherapy [6,7,8]. These results demonstrate the potential of conversion surgery as an effective treatment strategy for advanced gastric cancer, facilitated by contemporary chemotherapy. They also underscore the importance of identifying biomarkers that can predict chemotherapy resistance and treatment response to guide the optimal use of aggressive therapeutic strategies.

Tumors are composed of a heterogeneous mix of cells, some of which possess self-renewal capacity. Among these, cancer stem cells (CSCs), also known as tumor-initiating cells, are defined by their capacity for self-renewal and resistance to anticancer drugs [9,10,11,12]. CSCs are thought to play a central role in tumor initiation and progression. In addition to their contribution to chemoresistance, CSCs are also involved in the differentiation process, giving rise to a large number of non-tumorigenic cancer cells [13,14,15,16].

Owing to their critical roles in tumor relapse and metastasis, therapeutic strategies targeting CSCs have gained considerable interest. Several CSC-specific markers have been identified in gastric cancer, including cluster of differentiation 44 (CD44) and aldehyde dehydrogenase 1 (ALDH1) [17,18].

Investigating the relationship between the expression of these markers and treatment response holds promise for identifying such predictive biomarkers. However, studies examining the impact of CSCs on the histopathological effectiveness of chemotherapy in gastric cancer are limited.

CD44 is encoded by a single gene that comprises 19 exons. The first five and last five exons are conserved and form the standard isoform of CD44 (CD44s), which has a molecular weight of approximately 85–95 kDa. Variant isoforms (CD44v) arise through alternative splicing, incorporating combinations of nine variable exons with 10 constant exons [19,20,21,22,23,24]. These isoforms exhibit distinct biological activities. Notably, CD44 variant 9 (CD44v9) interacts with the xCT transporter on the cell membrane, facilitating cystine/glutamate exchange. This interaction enhances the intracellular synthesis of reduced glutathione, mitigates the accumulation of reactive oxygen species (ROS), and suppresses oxidative stress, thereby contributing to the maintenance of CSC properties, including self-renewal and chemoresistance [25].

In clinical studies, CD44v9 expression in gastric cancer tissues has been associated with chemotherapy efficacy and patient prognosis [13,26,27,28]. As evidence of chemotherapy resistance, Jogo et al. reported that among patients with stage II–III gastric cancer who received adjuvant chemotherapy after surgery, CD44v9-positive patients had a significantly poorer prognosis than CD44v9-negative patients. In contrast, in patients who did not receive adjuvant chemotherapy, CD44v9 expression did not have a significant impact on prognosis [29]. Additionally, in their study involving 185 individuals with stage I gastric cancer, Go et al. found that CD44v9 expression is an indicator of poor prognosis in early-stage gastric cancer but not in more advanced stages of the disease [27].

However, most current evidence is based on analyses of resected tumor specimens. Clinical data on the correlation between CD44v9 expression in pre-chemotherapy samples and histological response to chemotherapy remain limited. Clarifying this relationship would help validate, at the tissue level, previously reported associations between CD44v9 expression and anticancer drug efficacy observed at the cellular level. It also has significant clinical relevance, as CD44v9 may serve as a predictive marker to optimize treatment strategies and avoid unnecessary chemotherapy. This study aimed to clarify the relationship between CD44v9 expression in pretreatment endoscopic biopsy specimens and the histological response to chemotherapy in gastric cancer.

## 2. Materials and Methods

This single-center, retrospective observational analysis was conducted at the First Department of Surgery, University of Fukui, Fukui, Japan. The study identified 84 patients with pathologically confirmed gastric adenocarcinoma who underwent preoperative chemotherapy followed by surgical resection between January 2010 and December 2022. Patients were included if they had either (1) locally advanced, resectable gastric cancer (cT2-4 and/or N+ M0 disease) for which NAC was planned or (2) initially unresectable metastatic gastric cancer (M1 disease) treated with induction chemotherapy with the intent of conversion to surgery.

Conversion surgery cases were defined as those in which R0 resection became feasible after systemic therapy despite an initial diagnosis of unresectable disease. All 84 patients underwent gastrectomy with curative intent after completing preoperative chemotherapy. This study was approved by the Research Ethics Committee of the University of Fukui (Approval No. 20230017), and written informed consent was obtained from all participants before study initiation.

### 2.1. Chemotherapy and Surgical Treatment

Patients received combination chemotherapy regimens prior to surgery, with regimen selection based on contemporaneous clinical protocols and physician discretion. The docetaxel, cisplatin, and S-1 (DCS) triplet regimen was chosen as the first-line therapy due to reports of its high therapeutic efficacy. However, because severe adverse effects associated with DCS have also been reported, it was primarily administered to patients who were able to tolerate triplet chemotherapy [30,31,32]. In contrast, elderly patients or those with comorbidities received alternative treatments, such as platinum–fluoropyrimidine doublet regimens (e.g., S-1 plus cisplatin or capecitabine plus oxaliplatin), or other combinations in accordance with institutional practice. For analysis, patients were stratified by chemotherapy regimen into DCS and non-DCS groups to evaluate any differences in response or outcomes based on regimen intensity. Preoperative chemotherapy was typically administered for 2–4 cycles in initially resectable cases and continued until maximal tumor response (or up to 6–8 cycles, as clinically appropriate) in conversion therapy cases. After chemotherapy, patients underwent either subtotal or total gastrectomy with D2 lymphadenectomy (plus metastasectomy in conversion cases, when applicable) according to the Japanese gastric cancer treatment guidelines. Resected specimens were fixed in formalin and embedded in paraffin for pathological analysis.

### 2.2. RECIST Response Evaluation

The clinical outcomes of chemotherapy were assessed using the Response Evaluation Criteria in Solid Tumors (RECIST) version 1.0. This includes complete response (CR), where all known disease disappears; partial response (PR), characterized by a reduction of at least 50% in total tumor burden; stable disease (SD), defined as a decrease of less than 50% or an increase of less than 25%; and progressive disease (PD), marked by a tumor burden increase of ≥25% or the emergence of new lesions [33].

### 2.3. Pathological Response Evaluation

The histopathological response of the primary tumor to chemotherapy was assessed on the resected specimens by two independent staff pathologists who were blinded to all other clinical data. Evaluation was performed according to the tumor regression grading criteria of the Japanese Gastric Cancer Association [34]. A six-tier grading system (Grade 0–3, with subcategories a and b for Grades 1 and 2) was used to characterize the extent of residual viable tumor versus treatment-induced fibrosis or necrosis. A corresponding numerical response score (0–5) was recorded for each case. Grading was defined as follows:Grade 0 (score 0): no evidence of effect—no tumor cell death or regression, indicating complete chemotherapy resistance.Grade 1a (score 1): very slight effect—viable tumor cells occupy > 2/3 of the primary tumor area, indicating minimal tumor destruction by therapy.Grade 1b (score 2): moderate effect—viable tumor cells remain in >1/3 but <2/3 of the tumor, corresponding to a PR with roughly 1/3–2/3 of the tumor showing regression changes.Grade 2a (score 3): marked effect—only a small fraction of viable tumor (<1/3 of the tumor area) remains, with the majority (>2/3) of the tumor replaced by fibrosis or necrosis.Grade 2b (score 4): near-complete response—rare residual tumor cells present in the specimen (only microscopic foci of viable cancer cells), approaching a pathological CR.Grade 3 (score 5): complete response—no residual viable tumor cells detected in the primary site (pathological CR) [34].

For certain analyses, tumor regression results were dichotomized into poor vs. good responders. Patients with Grade 0 or 1a (score 0–1, indicating no or minimal response, considered chemotherapy-resistant) were classified as non-responders. Patients with Grade 1b or higher (score ≥ 2, indicating at least moderate response) were classified as responders.

### 2.4. CD44v9 Immunohistochemistry and Scoring

CD44v9 expression was evaluated by immunohistochemistry (IHC) using pretreatment endoscopic biopsy specimens. Biopsy tissue blocks that were fixed in formalin and embedded in paraffin were sliced into 2 μm thick sections, deparaffinized using xylene, and rehydrated through a series of ethanol solutions of increasing concentration. To reveal antigenic epitopes, the sections underwent antigen retrieval by autoclaving at 121 °C for 15 min. To prevent non-specific binding, the sections were treated with a diluted skim milk solution for 30 min at room temperature. Subsequently, they were exposed to a primary monoclonal antibody targeting CD44v9, which was developed in our laboratory, for 1 h at room temperature [35]. The antibody was diluted to approximately 0.1 μg/mL (~1:100) using an antibody-diluent (Thermo Fisher Scientific (Waltham, MA, USA)). After three washes with Tris-buffered saline (TBS), a secondary antibody was applied using the Histofine Simple Stain MAX PO detection system (Nichirei Biosciences Inc., Tokyo, Japan) for 30 min at room temperature. Subsequently, the sections were washed again with TBS, and the chromogenic reaction was developed using the Histofine DAB substrate kit (Nichirei Biosciences Inc.) for 4 min to visualize CD44v9 expression. Finally, the sections were counterstained with hematoxylin to stain the nuclei. Known CD44v9-positive gastric carcinoma tissues were used as positive controls, and staining with an isotype-matched irrelevant antibody was used as a negative control. Semi-quantitative scoring of CD44v9 immunoreactivity was performed by assessing the membranous staining intensity using a 0–3+ scoring system adapted from previously published criteria originally developed for CD44 IHC [22]. The intensity-based scoring guidelines were as follows: score 0 was assigned if there was negative or only weak membranous staining in <10% of tumor cells; score 1+, weak membranous staining in ≥10% of tumor cells or moderate intensity staining in <10% of cells; score 2+, moderate staining in ≥10% of tumor cells or strong (intense) staining in <10% of cells; and score 3+, intense membranous staining was observed in ≥10% of tumor cells (Figure 1). For each case, a consensus score (0, 1+, 2+, or 3+) was determined after reviewing any inter-observer differences (Figure 1). For analysis, CD44v9 expression was further categorized into low and high expressors: low CD44v9 expression was defined as an IHC score of 0, 1+, or 2+, whereas high CD44v9 expression was defined as a score of 3+. The evaluation of expression was conducted independently by two authors (K.S. and K.K.) under a blinded protocol, with the observers being unaware of the clinical outcomes and all other clinicopathological parameters of the patients.

### 2.5. Statistical Analysis

Associations among categorical variables were evaluated using the chi-square test. Multiple linear regression analyses were performed to identify factors associated with the CD44v9 IHC score and clinical factors. The CD44v9 model was adjusted for age, sex, tumor location, depth of invasion, lymph node metastasis, histological type, serum carcinoembryonic antigen (CEA) level, and distant metastasis. The pathological response model additionally included the chemotherapy regimen and CD44v9 expression status. Survival analyses were conducted to examine the prognostic value of pathological response and CD44v9 expression status. Disease-specific survival (DSS) was defined as the time from surgery to death attributable to gastric cancer, with patients alive at the most recent follow-up censored. The Kaplan–Meier method was employed to create survival curves, and the log-rank test was used to evaluate differences between groups. The Cox proportional hazards model was employed to calculate hazard ratios (HRs) and 95% confidence intervals (CIs). Statistical analyses were performed using IBM SPSS Statistics, version 21.0 (IBM Corp., Armonk, NY, USA). A *p*-value of less than 0.05 in a two-tailed test was deemed statistically significant.

## 3. Results

### 3.1. Associations Between CD44v9 Expression and Clinical Features

Table 1 presents the baseline demographic and clinical characteristics of the 84 patients with gastric cancer who underwent preoperative chemotherapy included in this study. Conversion surgery was performed in 44 patients (52.4%) with distant metastases, while 40 patients (47.6%) without distant metastases underwent radical resection following NAC. High CD44v9 expression (defined as immunohistochemical score 3) was observed in 21 patients (25%). The remaining 63 patients showed low CD44v9 expression: 39 patients scored 0, 7 scored 1, and 17 scored 2. In univariate analysis, tumor location showed a significant association, with high CD44v9 expression more frequent in tumors originating in the gastric fundus. No correlations were found with other clinical factors.

Multivariate linear regression analysis of the CD44v9 score and clinical factors confirmed that tumor location (regression coefficient [β], 0.306; 95% CI, 0.275–1.403; *p* = 0.004) and serum CEA level (β, 0.213; 95% CI, 0.020–1.190; *p* = 0.043) were independent predictors. Conversely, no significant association was found between CD44v9 expression and other clinical features (Table 2).

### 3.2. Clinical Responses Evaluation

Tumor response was assessed using RECIST version 1.0 [33].

The most frequently administered preoperative regimen was DCS, used in 63 out of 84 patients (75.0%). The remaining 21 patients (25.0%) received alternative cytotoxic combinations, including docetaxel and cisplatin (1 patient), docetaxel and S-1 (2 patients), 5-fluorouracil (5-FU) and oxaliplatin (4 patients), 5-FU and cisplatin (6 patients), and S-1 combined with intraperitoneal paclitaxel (8 patients). With DCS, PR was observed in 24 patients (38.1%), SD in 36 (57.1%), and PD in 3 (4.8%). In the non-DCS group, PR occurred in 12 patients (57.1%), SD in 8 (38.1%), and PD in 1 (4.8%). No CR was recorded. The expression intensity of CD44v9 showed no significant difference between patients with objective response and those without (*p* = 0.127) (Table 3).

### 3.3. Histological Responses Evaluation

Regarding histological treatment response, 3 patients were classified as grade 0, 28 as grade 1a, 20 as grade 1b, 18 as grade 2a, 5 as grade 2b, and 7 as grade 3. The histological response was evaluated as previously described and categorized as grade ≤1a vs. ≥1b. Patients with high CD44v9 expression showed a significantly poorer histological response than those with low expression (Table 4).

In the multivariate analysis of factors influencing histological response, high CD44v9 expression remained an independent predictor of poor response. No other clinical variables, including demographic or tumor characteristics, were independently associated with the histological response in this model (Table 5).

### 3.4. Relationship Between CD44v9 Expression Level and DSS

Kaplan–Meier analysis demonstrated that the intensity of CD44v9 expression was not prognostic: the 5-year overall survival rate was 46.3% for patients with a CD44v9 score ≤ 2 versus 63.8% for those with a CD44v9 score ≥ 3 (log-rank *p* = 0.177) (Figure 2). In the Cox proportional hazards model, only patients with distant metastasis identified before treatment, namely conversion cases, were associated with a poor prognosis (Table 6).

### 3.5. Relationship Between Histological Responses and DSS

Kaplan–Meier analysis demonstrated that histological response was prognostic: the 5-year overall survival rate was 38.5% for patients with a grade ≤ 1a response versus 58.1% for those with a grade ≥ 1b response (log-rank *p* = 0.045) (Figure 3). In the Cox proportional hazards model, patients with distant metastasis identified before treatment (i.e., conversion cases) and those with a poor pathological treatment response were associated with poor prognosis (Table 7).

## 4. Discussion

This study examined the association between CD44v9 expression in pretreatment endoscopic biopsy specimens and the histopathological efficacy of preoperative chemotherapy in patients with gastric cancer undergoing NAC or conversion surgery. CD44v9-positive tumors exhibited a significantly poorer histological response to anticancer agents. Although no definitive association was observed between CD44v9 expression and overall prognosis, patients who achieved a favorable histological response demonstrated superior survival outcomes. Clinicopathological analysis revealed that CD44v9 expression was significantly enriched in tumors located in the gastric fundus and in patients with elevated CEA levels (≥5 ng/mL). No correlations were observed with tumor depth, lymph node metastasis, or distant metastasis.

The cytotoxic effects of many anticancer drugs are mediated, in part, by oxidative stress [36,37,38]. CD44v9 facilitates cystine uptake via the xCT transporter, thereby promoting glutathione (GSH) biosynthesis, limiting intracellular ROS accumulation, and supporting the maintenance of CSCs properties [13,23]. These functions may underlie the reduced chemosensitivity observed in CD44v9-positive tumors. Consistent with this mechanism, Miyoshi et al. demonstrated that enforced CD44v9 expression increased intracellular GSH levels, suppressed ROS accumulation, and induced resistance to 5-fluorouracil (5-FU), whereas xCT inhibition enhanced 5-FU–induced cytotoxicity [39]. Similarly, Jogo et al. reported that silencing CD44v9 expression in vitro increased gastric cancer cell sensitivity to 5-FU, reinforcing the clinical relevance of CD44v9-mediated chemoresistance [29]. Furthermore, Ishimoto et al., using gastrointestinal cancer models, showed that suppression of CD44 variant expression reduced xCT levels and intracellular GSH. These alterations lead to an accumulation of ROS, which subsequently trigger the activation of p38 MAPK, a downstream effector of ROS, and induce the transcriptional upregulation of the cell cycle inhibitor gene p21 (CIP1/WAF1) [13,40,41].

These mechanisms may account for the association between CD44v9 expression in gastric cancer biopsy specimens and poor histopathological response to preoperative chemotherapy (including conversion therapy).

In a univariate analysis of 29 patients with gastric cancer who underwent NAC, Jogo et al. reported significantly lower histological response rates in patients with CD44v9-positive tumor cells in pretreatment biopsy specimens [29]. Our findings are consistent with this observation and further extend it by evaluating a larger patient cohort and confirming the association through both univariate and multivariate analyses.

In clinical applications, patients with tumors that are CD44v9-negative or have low CD44v9 expression can be expected to benefit from standard preoperative chemotherapy. This therapeutic effect may obviate extensive surgery and enable curative resection, even in patients with distant metastases. In contrast, in patients with high CD44v9 expression, standard 5-FU-based chemotherapy is more likely to be ineffective; therefore, surgical treatment may be considered for resectable cases, or the use of new anticancer drugs may be explored. For patients with CD44v-positive gastric cancer, a treatment strategy targeting the CD44v9–xCT–glutathione pathway has been proposed. A phase I clinical trial has already been conducted in which sulfasalazine, an xCT inhibitor, was added to a platinum-based chemotherapy regimen. This is an attempt to overcome chemotherapy resistance by inhibiting cystine uptake [42,43]. Wada et al. reported that the CD44v9–xCT–glutathione pathway is involved in chemoresistance in hepatocellular carcinoma. They also showed that inhibiting xCT can restore tumor sensitivity to cisplatin [44]. In addition, antibody–drug conjugate trials targeting CD44 variant 6 (CD44v6) and chimeric antigen receptor T-cell (CAR-T) therapies have been reported. Notably, CAR-T therapy has demonstrated antitumor efficacy in human multiple myeloma and acute myeloid leukemia [45,46,47]. With regard to CD44v9, not only has its role as a biomarker in gastric cancer been recognized, but it has also been suggested as a potential therapeutic target [48].

A correlation was observed between CD44v9 expression and histopathological therapeutic effects; however, no correlation was found with tumor response evaluated using RECIST. Wang et al. pointed out that RECIST may be unsuitable for lesions of gastrointestinal origin, as the assessment of residual tumor size can be distorted by post-chemotherapy tumor tissue fibrosis, necrosis, and edema [49]. Additionally, as this study included cases of conversion surgery, the therapeutic effect on the target lesions evaluated by RECIST may have differed from that on the primary tumor [2].

Sasaki et al. reported that the pathological therapeutic effect correlated with prognosis in patients who received DCS therapy as NAC for gastric cancer [50]. Lai et al. also reported that the pathological therapeutic effect correlated with prognosis in a study including 44 conversion cases and 111 NAC cases, and many other similar reports have been made [49,51,52,53]. The results of the present study are consistent with those of these previous reports, reaffirming that histopathological evaluation of treatment effects is an important predictor of postoperative prognosis.

In this study, a correlation was observed between CD44v9 expression in pretreatment biopsy tissue and the effectiveness of anticancer drug therapy, as well as between the effectiveness of anticancer drug therapy and prognosis. However, no relationship was observed between CD44v9 expression in biopsy samples and prognosis. Regarding the expression of CD44v9 and prognosis, Jogo et al. reported that positive CD44v9 expression in resected specimens is an independent poor prognostic factor for overall survival and relapse-free survival in patients with gastric cancer who had not received preoperative chemotherapy. In contrast, among 29 patients who underwent resection after preoperative chemotherapy, CD44v9 expression in pre-chemotherapy biopsy samples correlated with the histological therapeutic effect. However, the relationship between CD44v9 expression and prognosis has not been described in this cohort [29].

The following considerations can be made regarding the fact that the relationship between CD44v9 expression and prognosis could not be confirmed in this study. The effect of CD44v9 expression on prognosis is likely indirect, working through its association with poor treatment outcomes. In other words, tumors positive for CD44v9 may resist chemotherapy, leading to worse treatment outcomes. The residual presence of CD44v9 in resected specimens could play a role in later recurrence or progression. Moreover, CD44v9 expression reportedly changes during treatment. For example, Aso et al. showed that in head and neck squamous cell carcinoma, patients whose CD44v9 expression increased during preoperative chemoradiotherapy had poor prognoses. This suggests that post-treatment CD44v9 status may also influence the prognosis of gastric cancer [54].

Numerous studies have examined the expression of CD44v9 and its clinicopathological significance in gastric cancer. In a study involving 193 patients, Jogo et al. found that CD44v9 expression was notably linked to factors such as sex, invasion depth, lymphatic spread, vascular invasion, distant metastasis, and histological differentiation, with a higher occurrence in tumors that were well-to-moderately differentiated. In a study involving 103 patients, Yamakawa et al. found notable associations between CD44v9 expression and factors such as tumor size, invasion depth, lymph node metastasis, and tumor stage [29,55]. Go et al. reported no significant association between CD44v9 expression and tumor size, stage, or location in a study involving 333 patients. However, they found that CD44v9 expression was more frequently observed in well- to moderately differentiated tumors [27].

The discrepancies among these reports may be attributable to differences in the methods used to evaluate CD44v9 expression, as well as to variations in the patient populations. In the present study, CD44v9 expression was associated with tumor location and serum CEA levels; however, no significant associations were observed with other clinicopathological factors. Our cohort consisted exclusively of patients with advanced gastric cancer, many of whom exhibited deep invasion and extensive lymph node metastasis. These biases may have contributed to the differences in results compared with other reports.

However, biopsy-based evaluations are prone to sampling errors. Based on the results of the COMPASS trial and immunohistochemical analyses of gastric cancer biopsy specimens, Oshima et al. reported that selecting the most appropriate preoperative chemotherapy for each patient may be feasible. These findings support the utility of endoscopic biopsy for identifying predictive biomarkers [56,57].

This study has several limitations. First, the prognosis of gastric cancer is strongly influenced by other well-established prognostic factors, such as HER2 expression, microsatellite instability, and Epstein–Barr virus infection status. This study did not investigate the potential impact of these factors on treatment efficacy and clinical outcomes. Second, this study examined the association between pretreatment CD44v9 expression and treatment response and prognosis. However, dynamic changes in CD44v9 expression and the detailed categorization of chemotherapy regimens may also influence the therapeutic efficacy and clinical outcomes. Therefore, additional analyses of resected specimens and investigations using a single regimen could provide a more comprehensive evaluation. Third, in the survival analysis, patients who underwent conversion therapy and those who received NAC were not analyzed separately, which may have influenced the prognostic results. In the present study, this issue was addressed by including the presence or absence of distant metastasis at baseline as a covariate in the Cox regression model, thereby ensuring the validity of the analysis. Further studies with a larger sample size are warranted to enable subgroup analyses within each treatment group.

## 5. Conclusions

This study demonstrated that CD44v9 expression in gastric cancer biopsy specimens is a useful predictive marker for histological response to NAC. Moreover, patients who achieve a favorable histological response to NAC exhibit improved prognoses. Thus, assessing CD44v9 expression in gastric cancer biopsy specimens may inform treatment decision-making.

## Figures and Tables

**Figure 1 cancers-17-03657-f001:**
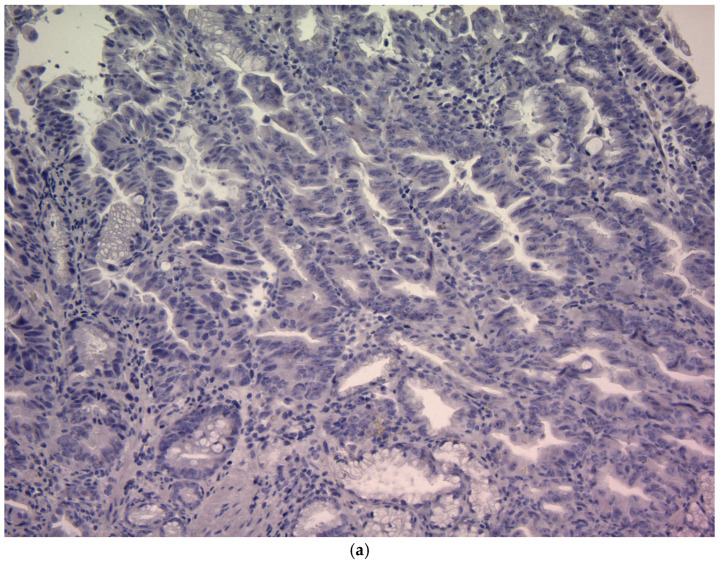
CD44v9 expression in gastric cancer cells as assessed by immunohistochemistry (×400). (**a**) Score 0: membranous staining in <10% of tumor cells; (**b**) Score 1+:weak membranous staining in ≥10% of tumor cells or moderate intensity staining in <10% of tumor cells; (**c**) Score 2+: moderate staining in ≥10% of tumor cells or strong (intense) staining in <10% of tumor cells; and (**d**) Score 3+: intense membranous staining is observed in ≥10% of tumor cells.

**Figure 2 cancers-17-03657-f002:**
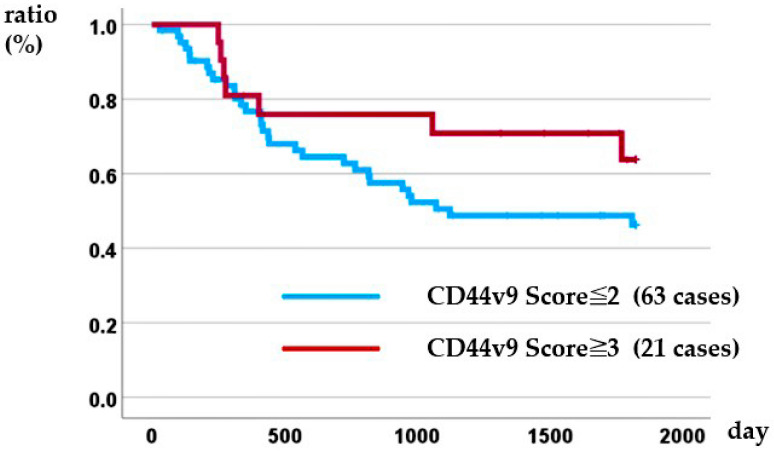
Relationship between CD44v9 expression and disease-specific survival.

**Figure 3 cancers-17-03657-f003:**
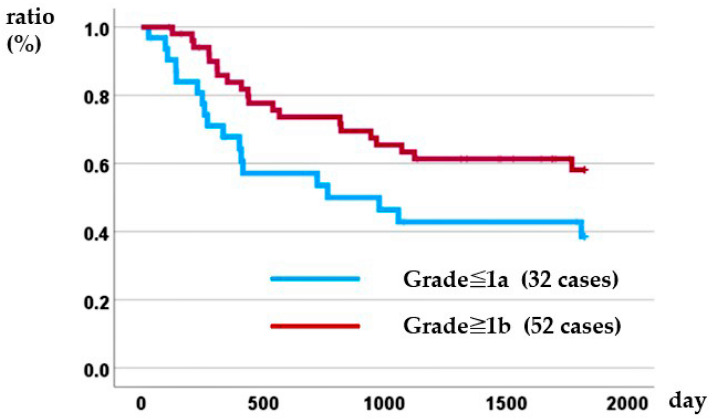
Relationship between histopathological therapeutic effect and disease-specific survival.

**Table 1 cancers-17-03657-t001:** CD44v9 expression in relation to clinical factors (before chemotherapy).

		No. of Cases	CD44v9 Score ≤ 2 Cases (%)	CD44v9 Score ≥ 3 Cases (%)	*p*-Value
All cases (%)		84	63 (75.0)	21 (25.0)	
Age	<60	22	19 (86.4)	3 (13.6)	0.124
	≥60	62	44 (71.0)	18 (29.0)	
Sex	Male	57	43 (75.4)	14 (24.6)	0.893
	Female	27	20 (74.1)	7 (25.9)	
Depth	≤T2	5	2 (40.0)	3 (60.0)	0.062
	≥T3	79	61 (77.2)	18 (22.8)	
Lymph node metastasis	Negative	17	14 (82.4)	3 (17.6)	0.433
	Positive	67	49 (71.3)	18 (28.7)	
Distant metastasis	Negative (NAC cases)	40	28 (70.0)	12 (30.0)	0.313
	Positive (Conversion cases)	44	35 (79.5)	9 (20.5)	
Location	Body, Antrum	58	49 (84.5)	9 (15.5)	0.003
	Fundus	26	14 (53.8)	12 (46.2)	
Histological type	undifferentiated	54	42 (77.8)	12 (22.2)	0.430
	differentiated	30	21 (70.0)	9 (30)	
CEA value	<5.0	61	49 (80.3)	12 (19.7)	0.066
	≥5.0	23	14 (60.9)	9 (39.1)	

CEA, carcinoembryonic antigen; NAC, neoadjuvant chemotherapy.

**Table 2 cancers-17-03657-t002:** CD44v9 expression scoring and clinical factors analyzed using multivariate analysis.

		Multivariate Linear Regression Analysis		
	Variables	β	95% CI	*p*-Value
Location	Body, antrum vs. fundus	0.306	0.275–1.403	0.004
CEA value	<5.0 vs. ≥5.0	0.213	0.020–1.190	0.043

β, regression coefficient; CI, confidence interval. (Adjusted for age, sex, depth, lymph node metastasis, distant metastasis, location, histological type, and CEA value).

**Table 3 cancers-17-03657-t003:** Relationship between CD44v9 expression and RECIST.

RECIST	CD44v9 Score ≤ 2 Cases	CD44v9 Score ≥ 3 Cases	Total
SD, PD	33	15	48
PR, CR	30	6	36

CR, complete response; PR, partial response; SD, stable disease; PD, progressive disease; RECIST, Response Evaluation Criteria in Solid Tumors.

**Table 4 cancers-17-03657-t004:** Histological responses and clinical factors.

		Therapeutic Effects Score ≤ 1 (%)	Therapeutic Effects Score ≥ 2 (%)	*p*-Value
Age	<60	9 (40.9)	13 (59.1)	0.752
	≥60	23 (37.1)	39 (62.9)	
Sex	Male	20 (35.1)	37 (64.9)	0.410
	Female	12 (44.4)	15 (55.6)	
Depth	≤T2	1 (20)	4 (80)	0.390
	≥T3	31 (39.2)	48 (60.8)	
Lymph node metastasis	Negative	6 (35.3)	11 (64.7)	0.790
	Positive	26 (38.8)	41 (61.2)	
Distant metastasis	Negative (NAC cases)	16 (40.0)	24 (60.0)	0.914
	Positive (conversion cases)	16 (36.4)	28 (63.6)	
Location	Body, antrum	22 (37.9)	36 (62.1)	0.963
	Fundus	10 (38.5)	16 (61.5)	
Histological type	Undifferentiated	20 (37.0)	34 (63.0)	0.789
	Differentiated	12 (40.0)	18 (60.0)	
CEA	<5.0	23 (37.7)	38 (62.3)	0.905
	≥5.0	9 (39.1)	14 (60.9)	
Chemotherapy type	DCS	22 (34.9)	41 (65.1)	0.299
	Non DCS	10 (47.6)	11 (52.4)	
CD44v9	Score ≤ 2	19 (30.2)	44 (69.8)	0.009
	Score ≥ 3	13 (61.9)	8 (38.1)	

DCS, docetaxel, cisplatin, and S-1.

**Table 5 cancers-17-03657-t005:** Histological therapeutic scoring and clinical factors analyzed using multivariate analysis.

		Multivariate Linear Regression Analysis		
	Variables	β	(95% CI)	*p*-Value
CD44v9	Score ≤ 2 vs. score ≥ 3	−0.218	(−1.384 to −0.013)	0.046

(Adjusted for age, sex, depth, lymph node metastasis, distant metastasis, location, histological type, CEA value, chemotherapy type, and CD44v9 expression).

**Table 6 cancers-17-03657-t006:** Multivariate Cox analysis for DSS incorporating CD44v9 expression.

			Multivariate	
	Variable	HR	95% CI	*p*-Value
Age	<60 vs. ≥60	0.503	0.244–1.037	0.063
Sex	Male vs. Female	0.749	0.363–1.546	0.435
Distant metastasis	NAC cases vs. Conversion cases	2.269	1.133–4.544	0.021
CD44v9 expression	Score ≤ 2 vs. score ≥ 3	0.704	0.301–1.645	0.418

HR, hazard ratio.

**Table 7 cancers-17-03657-t007:** Multivariate Cox analysis for DSS incorporating histological responses.

			Multivariate	
	Variable	HR	95% CI	*p*-Value
Age	<60 vs. ≥60	0.439	0.216–0.891	0.023
Sex	Male vs. Female	0.726	0.352–1.498	0.386
Distant metastasis	NAC vs. conversion cases	2.415	1.220–4.778	0.011
Histological responses	Grade ≤ 1a vs. Grade ≥ 1b	0.460	0.242–0.875	0.018

## Data Availability

All data included in this study are available upon request from the corresponding author.

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
