# Peer review of "CD44v9 Expression in Pretreatment Biopsies as a Predictor of Chemotherapy Resistance in Gastric Cancer"

_cancers, 2025, doi:10.3390/cancers17223657_

Round 1
Reviewer 1 Report
Comments and Suggestions for Authors
The work looks great, clear, and well referenced. However, I have a few suggestions below to enhance the paper further.
A brief clarification of the chemotherapy regimens heterogeneity and its potential influence on treatment response would be beneficial to add. The methodology looks great as well as detailed including the immunohistochemical protocol and grading criteria. However, clarify the Antibody source wherever needed by providing the manufacturer, clone, and catalog number to ensure reproducibility. Also, explain Regimen Selection since different chemotherapy regimens (DCS vs. others) were used to discuss whether regimen type could confound CD44v9 response associations. Tables and figures are well structured. The statistical reporting is transparent. Figures 2 & 3 effectively convey survival outcomes. Adding sample numbers to Kaplan–Meier plots would maybe enhance clarity. The conclusions accurately reflect the findings: CD44v9 high expression correlates with poor histological response but not directly with survival, implying an indirect prognostic role through chemoresistance. It would be benefical to add Clinical Context. For example, briefly discuss how CD44v9 testing could be implemented in clinical decision-making or future prospective trials etc etc. The limitations are fairly acknowledged. Lastly, a short language proofread and a few minor grammatical corrections would enhance clarity. Overall, this is a well conducted as well as clinically relevant study that provides useful evidence supporting CD44v9 as a predictive biomarker for chemotherapy resistance in gastric cancer.
Author Response
EDITOR’S SPECIFIC COMMENTS:
- A brief clarification of the chemotherapy regimens heterogeneity and its potential influence on treatment response would be beneficial to add.
(Response)
We appreciate these helpful suggestions.
First, we have provided a more detailed description of the chemotherapy regimens used. We added the following information regarding the chemotherapy content to Section 3.2, Clinical Response Evaluation, in the Results:
“The most frequently administered preoperative regimen was DCS, used in 63 out of 84 patients (75.0%). The remaining 21 patients (25.0%) received alternative cytotoxic combinations, including docetaxel and cisplatin (1 patient), docetaxel and S-1 (2 patients), 5-fluorouracil (5-FU) and oxaliplatin (4 patients), 5-FU and cisplatin (6 patients), and S-1 combined with intraperitoneal paclitaxel (8 patients).”
The added section is highlighted in yellow.
As pointed out, we considered the possibility that differences in chemotherapy regimens might influence treatment efficacy. Therefore, in our analysis of histological treatment responses, we divided patients into DCS and non-DCS groups. In univariate analysis, the regimen type did not affect treatment response. We also included this classification as a covariate in the multivariate model. After adjustment, CD44v9 remained an independent predictor of poor histological response, and the regimen type was confirmed not to influence histological treatment outcomes (as described in Section 3.3, Histological Responses Evaluation).
However, because detailed categorization of chemotherapy regimens may potentially affect treatment outcomes, we revised the limitations in the Discussion as follows:
“However, dynamic changes in CD44v9 expression and the detailed categorization of chemotherapy regimens may also influence the therapeutic efficacy and clinical outcomes. Therefore, additional analyses of resected specimens and investigations using a single regimen could provide a more comprehensive evaluation.”
The added text is highlighted in yellow.
EDITOR’S SPECIFIC COMMENTS:
- clarify the Antibody source wherever needed by providing the manufacturer, clone, and catalog number to ensure reproducibility.
(Response)
We appreciate these helpful suggestions.
The primary anti-CD44v9 antibody used for immunohistochemistry was a mouse monoclonal antibody developed in our laboratory. Details regarding its preparation are described in the following publication, which we have now added as Reference 35:
Seki K, Yamaguchi A, Goi T, Nakagawara G, Matsukawa S, Urano T, Furukawa K.
Inhibition of liver metastasis formation by anti-CD44 variant exon 9 monoclonal antibody. International Journal of Oncology. 1997 Dec;11(6):1257-61.
In the “Preparation of Antibody” section of this paper, the following method is described:
“A mouse was immunized subcutaneously (s.c.) three times with CD44v8-10 fusion protein at two-week intervals: the first time with 50 μg of protein with complete Freund's adjuvant, the second with 100 μg of protein with incomplete Freund's adjuvant, and the third with 100 μg of protein alone. Spleen cells were fused with the murine myeloma cell line NS-1. The hybridoma culture supernatants were assayed for reactivity with CD44v8-10 protein using an enzyme-linked immunosorbent assay (ELISA) and immunoblotting. Positive cultures were cloned by limiting dilution three times to obtain the mAb 44-IV. The mAb 44-IV recognizes an epitope on exon v9 of CD44. This antibody was purified by ammonium sulfate precipitation and protein G-Sepharose chromatography (Pharmacia, Uppsala, Sweden), and then dialyzed overnight at 4°C against 6 L of phosphate-buffered saline (PBS).”
EDITOR’S SPECIFIC COMMENTS:
- explain Regimen Selection since different chemotherapy regimens (DCS vs. others) were used to discuss whether regimen type could confound CD44v9 response associations.
(Response)
We appreciate these helpful suggestions.
We have revised the text in the Materials and Methods section, specifically under Chemotherapy and Surgical Treatment, as indicated below. Additionally, we have included references 30, 31, and 32 concerning the DCS regimen. The added text has been highlighted in yellow.
Patients received combination chemotherapy regimens prior to surgery, with regimen selection based on contemporaneous clinical protocols and physician discretion. The docetaxel, cisplatin, and S-1 (DCS) triplet regimen was chosen as the first-line therapy due to reports of its high therapeutic efficacy. However, because severe adverse effects associated with DCS have also been reported, it was primarily administered to patients who were able to tolerate triplet chemotherapy. In contrast, elderly patients or those with comorbidities received alternative treatments, such as platinum–fluoropyrimidine doublet regimens (e.g., S-1 plus cisplatin or capecitabine plus oxaliplatin), or other combinations in accordance with institutional practice.
EDITOR’S SPECIFIC COMMENTS:
- We appreciate the suggestion to enhance clarity by adding the number of patients at risk to Figures 2 and 3.
(Response)
We appreciate these helpful suggestions.
In the revised figures, we have included the number of patients remaining in each group at the beginning of the survival curves.
EDITOR’S SPECIFIC COMMENTS:
- It would be benefical to add Clinical Context.
(Response)
We appreciate these helpful suggestions.
We have added the following text to the Discussion section. References 45, 46, 47, and 48 have also been included accordingly. In addition, the abbreviations CD44v6 and CAR-T have been added to the list of abbreviations. The added text has been highlighted in yellow.
In addition, antibody–drug conjugate trials targeting CD44 variant 6 (CD44v6) and chimeric antigen receptor T-cell (CAR-T) therapies have been reported. Notably, CAR-T therapy has demonstrated antitumor efficacy in human multiple myeloma and acute myeloid leukemia. With regard to CD44v9, not only has its role as a biomarker in gastric cancer been recognized, but it has also been suggested as a potential therapeutic target.
Reviewer 2 Report
Comments and Suggestions for Authors
What is the rationale for categorizing IHC scores 0, 1+, and 2+ as a unified "low expression" category? This appears to disregard potentially vital information. Have the authors contemplated examining the IHC scores as a continuous variable or employing an alternative cutoff point? A detailed investigation may uncover a more robust and direct association with both histology response and prognosis.
The study illustrates that although CD44v9 expression forecasts a negative histology response, it does not have a direct correlation with prognosis. In what manner do the authors elucidate this discrepancy? If a biomarker indicates a suboptimal cellular response, why does this not correlate with a deteriorated clinical outcome for the patient? Could the authors elucidate the potential confounding variables or alternative biological mechanisms that may be affecting patient survival, independent of the initial histological response?
What is the rationale for categorizing IHC scores 0, 1+, and 2+ as a singular "low expression" group? This looks to discard potentially valuable data. Have the authors contemplated examining the IHC scores as a continuous variable or employing an alternative cutoff point? A detailed investigation may uncover a more robust and direct association with both histology response and prognosis.
Given the significant clinical differences between the NAC and conversion therapy cohorts, is it appropriate to analyze them as a single group? Could the lack of a significant association between CD44v9 and survival be due to the confounding effect of lumping these two disparate groups together? A subgroup analysis for each cohort is warranted to clarify the predictive value of CD44v9 in these distinct clinical scenarios.
What is the rationale for categorizing IHC scores 0, 1+, and 2+ as a singular "low expression" group? This looks to discard potentially valuable data. Have the authors contemplated examining the IHC scores as a continuous variable or employing an alternative cutoff point? A detailed investigation may uncover a more robust and direct association with both histology response and prognosis.
The report identifies a difference between the RECIST assessment and the pathological reaction. This is a significant point that warrants further examination in the discourse. Could the authors clarify why RECIST may be an inadequate metric for assessing response in this situation and how this could influence the interpretation of other research that depend exclusively on RECIST?
Author Response
EDITOR’S SPECIFIC COMMENTS:
1.What is the rationale for categorizing IHC scores of 0, 1+, and 2+ as a unified "low expression" category? This appears to disregard potentially vital information. Have the authors contemplated examining the IHC scores as a continuous variable or employing an alternative cutoff point? A detailed investigation may uncover a more robust and direct association with both histology response and prognosis.
(Response)
We appreciate your comments and the opportunity to clarify this important point.
In our study, we categorized IHC scores of 0, 1+, and 2+ as a unified “low expression” group based on the scoring system proposed by Hirata et al. (Reference 22 in our manuscript; see page 5, line 199. In their study, this grouping was also applied, defining only score 3+ as “high expression.” We followed this precedent to maintain consistency with the established literature and create a simplified and clinically interpretable classification by distinguishing cases with strong expression from all others.
We acknowledge that alternative methods exist, including scoring systems that combine staining proportion and intensity. However, since our study was based on biopsy specimens, variability in the sampling site and tumor heterogeneity may have affected the staining proportions. Therefore, we adopted an intensity-based system that we considered to be more appropriate for evaluating small biopsy samples.
It is worth noting that in our multivariate analysis of factors associated with CD44v9 expression, we treated IHC scores as continuous variables to allow for linear regression modeling. This approach enabled us to examine multiple factors influencing CD44v9 expression, which would not have been possible if we had used dichotomized groups from the outset, as this would have necessitated logistic regression and limited the number of variables that we could assess simultaneously.
Conversely, for analyses related to treatment response and prognosis, we used the high vs. low classification, following the same rationale as Hirata et al. (Ref. 22). To ensure consistency across analyses, we dichotomized other variables, such as age, depth of invasion, tumor location, serum CEA level, and chemotherapy regimen.
We agree that further investigation into the significance of weak or moderate CD44v9 staining is warranted. In future studies, we intend to explore the potential prognostic or predictive values of these intermediate expression categories.
Once again, we thank the reviewer for raising this important issue, which has helped us refine the manuscript and clarify our methodological rationale.
EDITOR’S SPECIFIC COMMENTS:
2.The study illustrates that although CD44v9 expression forecasts a negative histology response, it does not have a direct correlation with prognosis. In what manner do the authors elucidate this discrepancy? If a biomarker indicates a suboptimal cellular response, why does this not correlate with a deteriorated clinical outcome for the patient? Could the authors elucidate the potential confounding variables or alternative biological mechanisms that may be affecting patient survival, independent of the initial histological response?
(Response)
We appreciate the reviewer’s suggestion.
Several studies, including Reference 29, have reported that CD44v9 expression in resected specimens is associated with a poor prognosis. Since resected specimens allow evaluation of the entire tumor, CD44v9 expression in these samples may more accurately reflect the tumor’s biological aggressiveness and thus be more reliable for prognostic prediction.
In Reference 46, Aso et al. studied head and neck squamous cell carcinoma and found that CD44v9 expression in pretreatment biopsies did not correlate directly with survival. However, in patients with poor response to chemoradiotherapy, the emergence of CD44v9-positive cells after treatment was significantly associated with a worse prognosis. This suggests that therapy may induce CD44v9-positive cancer stem-like cells and that dynamic changes in CD44v9 expression may be more prognostically relevant than baseline expression alone.
In our study, CD44v9 positivity was associated with a poor histological response (Table 4). However, we did not observe a significant association with overall survival. One potential explanation for this discrepancy is that CD44v9 expression may increase during treatment. As we did not evaluate CD44v9 expression in post-treatment resected specimens, we could not assess this change. We plan to investigate this in future studies.
Additionally, differences in postoperative treatments such as adjuvant chemotherapy may have influenced survival outcomes, potentially masking the effect of baseline CD44v9 expression.
We have described this dissociation between histologic response and prognosis in the manuscript (page 12, lines 389–394).
EDITOR’S SPECIFIC COMMENTS:
3.Given the significant clinical differences between the NAC and conversion therapy cohorts, is it appropriate to analyze them as a single group? Could the lack of a significant association between CD44v9 and survival be due to the confounding effect of lumping these two disparate groups together? A subgroup analysis of each cohort is warranted to clarify the predictive value of CD44v9 in these distinct clinical scenarios.
(Response)
We appreciate the reviewer’s thoughtful comment.
The primary objective of this study was to evaluate the relationship between CD44v9 expression and response to preoperative chemotherapy. Therefore, we combined both NAC and conversion therapy cases as a single cohort, as both groups had undergone preoperative treatment. If we had limited the analysis to either NAC or conversion therapy cases alone, the sample size for each group would have been approximately 40, resulting in insufficient statistical power. Therefore, we chose to analyze them together to ensure adequate statistical robustness.
However, we fully recognize that the NAC and conversion cohorts differ in their clinical characteristics. To address this potential confounding factor, we included “treatment purpose” (i.e., NAC vs. conversion) as an adjustment variable in our multivariate survival analysis. This approach was intended to minimize the influence of heterogeneity on the prognostic analysis.
Regarding subgroup analysis, while the limited number of cases in each group constrained the statistical feasibility of a fully powered subgroup comparison, we conducted exploratory subgroup evaluations within the NAC and conversion therapy groups to examine whether any consistent trends could be observed.
In the NAC group (n = 40), the 5-year survival rate was 75.0% in cases with high CD44v9 expression and 60.2% in other cases, with no significant difference in prognosis (P = 0.502). Additionally, cases with a histological treatment effect grade of ≥1b had a 5-year survival rate of 76.2%, compared to 50.0% in other cases, but the difference was not statistically significant (P = 0.502).
In the conversion therapy group (n = 44), the 5-year survival rate was 48.6% in cases with high CD44v9 expression and 34.9% in other cases, again with no significant prognostic difference (P = 0.353). Similarly, cases with a histological treatment effect grade of ≥1b had a 5-year survival rate of 43.8%, versus 23.0% in other cases, although this difference was not statistically significant (P = 0.177).
These results are available upon request and may be added as supplementary material if the reviewer finds them helpful. We also plan to expand our cohort in future studies to enable a more robust stratified analyses.
We have described this limitation in the Discussion section on page 12, lines 427–432.
EDITOR’S SPECIFIC COMMENTS:
4.The report identifies a difference between the RECIST assessment and the pathological reaction. This is a significant point that warrants further examination in the discourse. Could the authors clarify why RECIST may be an inadequate metric for assessing response in this situation and how this could influence the interpretation of other research that depend exclusively on RECIST?
(Response)
We appreciate the reviewer’s insightful comments.
As described in the manuscript (page 11, lines 360–366), our study revealed a discrepancy between the RECIST-based radiologic assessment and pathological response. This issue has also been highlighted by Wang et al. (Reference 49), who noted that chemotherapy can induce tissue changes, such as fibrosis, necrosis, and edema, which may distort the measurement of residual tumor size. Consequently, RECIST may not accurately reflect the treatment response in gastrointestinal malignancies, as tumor shrinkage does not necessarily correlate with the degree of tumor cell viability or pathological regression.
Furthermore, our study included conversion therapy cases in which the target lesion evaluated by RECIST was sometimes different from the primary tumor. In such cases, the radiological response assessed using RECIST may not correspond to the therapeutic effect on the primary tumor.
Given these limitations, reliance on RECIST alone may lead to misinterpretation of therapeutic efficacy in both clinical practice and research settings. Studies that use RECIST as the sole metric should consider the potential discordance with pathological response, especially in gastrointestinal cancer settings.
​​
Round 2
Reviewer 2 Report
Comments and Suggestions for Authors
The author responded to my comments